# Extensive Changes in Transcription Dynamics Reflected on Alternative Splicing Events in Systemic Lupus Erythematosus Patients

**DOI:** 10.3390/genes12081260

**Published:** 2021-08-18

**Authors:** Sofia Papanikolaou, George K. Bertsias, Christoforos Nikolaou

**Affiliations:** 1School of Medicine, University of Crete, Voutes, 70013 Heraklion, Greece; papanikolaou.sofia@hotmail.com (S.P.); gbertsias@uoc.gr (G.K.B.); 2Biomedical Sciences Research Center “Alexander Fleming”, Institute of Bioinnovation, Fleming 14-16, 16672 Athens, Greece; 3IMBB, FORTH, Voutes, 70013 Heraklion, Greece

**Keywords:** alternative splicing, systemic lupus erythematosus, transcription

## Abstract

In addition to increasing the complexity of the transcriptional output, alternative RNA splicing can lead to the reduction of mRNA translation or the production of non-functional or malfunctional proteins, thus representing a vital component of the gene regulation process. Herein, we set out to detect and characterize alternative splicing events that occur in whole-blood samples of patients with Systemic Lupus Erythematosus (SLE) as compared to healthy counterparts. Through the implementation of a computational pipeline on published RNA-sequencing data, we identified extensive changes in the transcription dynamics affecting a large number of genes. We found a predominance of intron retention events, with the majority introducing premature stop codons, suggestive of gene repression, in both inactive and active SLE patient samples. Alternative splicing affected a distinct set of genes from the ones detected as differentially expressed in the same comparisons, while alternatively spliced genes tended to reside in genome areas associated with increased gene co-expression. Functional analysis of genes affected by alternative splicing pointed towards particular functions related to metabolism and histone acetylation as of potential interest. Together, our findings underline the importance of incorporating alternative splicing analyses in the context of molecular characterization of complex diseases such as SLE.

## 1. Introduction

Pre-mRNA splicing is a post-transcriptional mechanism by which introns are removed and exons are joined together leading to mature mRNA molecule formation. Different coding regions can be retained or not in the final transcript, a process called alternative splicing (AS) that generates novel mRNA isoforms. This procedure results in transcriptome and proteome diversity, as a gene can produce multiple transcripts. It is estimated that almost 95% of genes with multiple exons are subject to AS [1] and a gene can express approximately 10–12 unique isoforms [2].

It has been revealed that the most frequent event of AS in metazoans is skipping of a cassette exon, usually containing long neighboring introns [3]. Another type is intron retention (IR), when introns are included in the final transcript [4]. Other events consist of alternative usages of 5′ donor splice sites or 3′ acceptor splice sites within exons. It is estimated that the distance between constitutive and alternative splice site is about 2–12 nucleotides [5]. In addition, there are mutually exclusive exons, where the inclusion of an exon prevents the inclusion of another [6]. 

As the vast majority of the affected sites of mRNA belong to open reading frames (ORFs), they result in their disruption or the introduction of premature termination codons (PTC). These changes may impact the structure and the function of the produced proteins. Splice variants containing a PTC can be subject to elimination via nonsense mediated mRNA decay (NMD), a mechanism that reduces errors born during transcription by preventing translation of aberrant mRNA molecules [7]. The decision as to whether a splice variant that carries a PTC will have this fate, depends on the existence of an exon–exon junction over 50 nucleotides downstream of the PTC [8,9,10]. This procedure leads to mRNA destruction. Alternatively, mRNA translation will take place and be completed, after the ribosome reaches the premature termination codon. AS can also introduce variability in non-coding regions, regulating mRNA stability, subcellular localization and translation efficiency. Among the different types of AS events, IR has been recently proposed to act as an additional layer of transcriptional “tuning” not producing alternative protein products but suppressing inappropriately expressed transcripts [11,12,13]. Thus, AS may be considered a major component of the transcriptional regulation process, affecting both homeostasis and disease.

In Systemic Lupus Erythematosus (SLE), a complex autoimmune disorder characterized by production of autoantibodies against nuclear and cytoplasmic antigens and multisystem inflammation, previous studies have revealed that disease implicated genes are subject to AS. Several AS events have been reported to affect immune signaling genes including exon skipping in the mRNA of the B-cell scaffold protein with ankyrin repeats (*BANK1*) and an alternative 3′ splice site of the leukocyte immunoglobulin like receptor A1 (*LILRA2*). Both events lead to the production of a novel isoform that may lack a binding region [14,15]. Moreover, splicing QTLs (sQTLs), i.e., DNA sequences linked with variation in splicing isoform abundance, that are related to SLE, have been identified [16]. Genes containing sQTLs include *TCF7*, *SKP1, BLK, NADSYN1, IKZF2, WDFY4* and *IRF5*. In a recently conducted transcriptome analysis of RNA, we collected whole-blood sequencing data from 142 SLE patients and 58 healthy individuals [17]. A functional assessment of differentially expressed (DE) genes showed enrichment in immune signaling pathways like p53, IFN and NOD-like receptor and molecular processes like RNA transport. A rudimentary quantification of alternative splicing events revealed perturbed mRNA splicing of SLE patients and splicing QTL analysis indicated a set of spliced and DE genes having sQTL too. The alternatively spliced genes mediate immune system and type I interferon signaling pathways.

In this study, we perform a detailed and analytical study of splicing dynamics in this extensive dataset. We focus on the computational identification of AS events between healthy individuals and patients suffering from SLE with varying degrees of disease activity. We explore the different modes of transcriptional deregulation through both DE and AS and put forward the hypothesis that AS may be tuning the transcriptome through increased intron retention events. Importantly, we identify AS-affected genes to be enriched in a distinct subset of cellular functions, not directly related to inflammation and immunity. Our findings suggest that AS constitutes an additional layer of transcriptional regulation in SLE and might affect a number of important biological pathways not previously detected by differential gene expression analysis. 

## 2. Materials and Methods

### 2.1. SLE RNASeq Data Integration and Patient Stratification

RNA sequencing data were obtained from whole blood samples of 58 healthy individuals, 46 SLE patients at inactive disease state and 79 patients at active state. Data was produced and initially sequenced as described in Panousis et al. [17]. The categorization of the patients into different disease activity states was accomplished by assessing their clinical SLE Disease Activity Index (SLEDAI-2K), an index of disease severity, which is calculated on the basis of a number of disease descriptors. In general, patients with a clinical SLEDAI-2K of 0 at the time of sampling were qualified as being in the inactive state, whereas those with clinical SLEDAI-2K ≥ 5 were deemed as being in the active state. However, the final assignment to a disease stage was given by the clinician responsible [17].

### 2.2. Differential Splicing Analysis

The detection of AS events was conducted using Vertebrate Alternative Splicing and Transcription Tools (Vast-tools) [18]. The analysis pipeline was implemented through three main steps. 

The first was the alignment of input files, where reads were mapped against hg19 human reference genome and subsequently to predefined splice junction libraries. hg19 was chosen in order to allow for more immediate cross-comparison with our previous findings related to the expression dynamics of the same dataset [17]. Read counting was performed at gene level. Due to the low read coverage (total read depth) of the replicates of each disease state group (~10 million reads/sample), we randomly merged the samples into multiple subgroups, so that each subgroup contained approximately 10 samples (Table 1).

Second, all output files were joined together into a table containing information about each predefined AS event, through the “combine” module. The provided information included symbol of the affected gene, event ID (interconnected with VastDB) [19], coordinates and length of the region spanning the AS event, as well as the full set of coordinates, event type, percent spliced in and quality scores. Examination of the differential usage of alternative splicing events under active or inactive disease state in contrast to healthy was performed via Bayesian inference [20].

Splicing is assessed via the percentage of reads supporting the inclusion of the examined event in a transcript, coined as Percent Spliced In (PSI). We considered a splicing event to be statistically significant, if the expected value of the difference of Percent Spliced In (dPSI) between two groups was greater than the minimum (non-zero) value of dPSI at 0.95 probability of acceptance. These events were thought to be relevant but as their changes in dPSI were modest, we further filtered our results based on the minimum fraction of samples in which the event has adequate read coverage (Appendix A). The specified fraction was 20%. In all, we used rather stringent criteria for both read coverage and dPSI significance, in order to make sure that the reported events were highly reliable. 

### 2.3. Prediction of Nonsense Mediated Decay

To examine the introduction of PTC that could lead either to nonsense mediated decay or termination of translation, we retrieved retained intron sequences, detected the premature termination codon and computed the distance between the last exon–exon junction. Stop codons positioned in distance of at least 50 bases from downstream exon-exon junctions could elicit nonsense mediated decay, while stop codons positioned no less than 50 bases could terminate translation.

### 2.4. Differential Gene Expression Analysis

The gene counts obtained from the first step were used as input for differential expression analysis with DESeq2 [21]. Thresholds for absolute log2-fold-change greater than 1 at a 5% false discovery rate were imposed, in order to detect the statistically significant differentially expressed (DE) genes (Appendix A). Functional enrichment analysis was performed using gProfileR [22] and Cluster Profiler [23].

## 3. Results

### 3.1. Extensive Perturbation of Splicing and Predominance of Intron Retention Events

We observed an extensive perturbation of AS when comparing healthy individuals against SLE patients at different disease states. The total number of statistically significant splicing events was 371 in the comparison between SLE inactive and healthy individuals and 578 in SLE active versus healthy, while the affected genes were 317 and 500 correspondingly. We found a predominance of intron retention events since in both comparisons the majority of AS events were retained introns, followed by exon skipping (Figure 1). In terms of event sequence size, there were no significant differences between active and inactive disease states, as the median lengths of the retained introns were 1138.5 and 1404 bp, respectively (Appendix A). Similarly, the median lengths of the skipped exons were 108 and 118.5 bp for the inactive-vs.-healthy and the active-vs.-healthy comparisons, respectively (Appendix A). Skipped exon sizes are close to the genome average but retained introns are overall shorter than the average human intron (~6kbp). As shorter introns are generally easier to be defined and spliced out by the splicing machinery, their enrichment in the intron retention list is likely reflecting functional tendencies. Intron retention was the most prominent AS event type at gene level too, with 183 out of a total of 317 genes in the inactive state being affected by at least one such event. The corresponding numbers, for the active state were 318 genes out of 500. While, for most of these genes we observe only a single event, certain genes were found to undergo intron retention at multiple sites. Notably, the *ACO1* gene, encoding for an aconitase, has 12 retained introns in the inactive and six in the active SLE patient samples. Retained intron genes were further examined for the introduction of premature termination codon and their potential elimination via NMD. The prediction of transcripts leading to NMD, were performed for the genes with retained introns in either of the two comparisons. The percentage of the retained intron transcripts that can be subjected to elimination via NMD was 0.938 in the comparison of SLE active versus Healthy and 0.967 in comparison of SLE inactive versus Healthy.

### 3.2. Relationship between Differentially Expressed Genes and Alternative Splicing Events

#### 3.2.1. Alternative Splicing and Differential Expression Involve Different Genes

Τhe increased number of AS events in the SLE active state is in agreement with data from differential expression, in which active SLE patients show a greater number and more intense changes at transcript levels [17]. We examined whether AS events occur among the subset of genes that are differentially expressed (DE) at both active and inactive disease states. We compared the genes that were DE in the active and inactive states with genes undergoing AS (Figure 2). Even though genes affected by AS were significantly more than DE ones, their overlap was very small for both inactive and active states. Only 2.5% of active state DE genes were also involved in AS, while the corresponding percentage for the inactive state was 1.5%. This suggests that AS affects a different subset of genes, than those whose expression changes and that, overall, splicing regulation is likely taking place at a qualitative and not quantitative level, meaning that it is directed against a different subset of genes.

AS also appears to be more dynamic than DE, since AS events are less consistent between active and inactive disease states. When comparing AS and DE genes between the two states, we found that DE genes between active and inactive SLE were common to a much greater extent (175/486 genes, Jaccard Index = 0.36) than genes affected by AS (54/763 events, Jaccard Index = 0.07). Breaking down the genes affected by AS events that were common between the two disease states for different types of events, showed no significant differences with 6/101 for alternative donor/acceptor, 8/230 for exon skipping and 27/474 for intron retention. The dissimilarity of genes affected by AS compared to DE suggests that transcriptional modulation by splicing is a much more complex process compared to the production of mature mRNA captured by standard RNASeq approaches. The few cases where splicing was consistent between active and inactive SLE states were 54 (although there were only 14 splicing events shared in both comparisons) (Figure 3). Interestingly, the dPSI values of most of these cases were increased in active samples.

We next examined the gene expression level changes of genes affected by AS in search of particular trends in gene expression values (Figure 4). We found that only a small number of events (4 for inactive and 12 for active SLE patient samples) were also differentially expressed in the same conditions. With the exception of one, all are over-expressed. Even though this is not statistically significant (chi-square test, *p* = 0.07), it may be representative of an interesting trend for AS to operate in genes with increased mRNA transcripts. More interestingly, the majority of these cases are affected by intron retention events. Taken together, these results suggest a rather complex, compensatory mode of gene regulation, with albeit small fraction of genes, being repressed through AS, while being over-expressed at the mRNA level. Together, these observations point towards the need for more nuanced readouts of RNASeq output, that account for splicing dynamics next to mRNA levels. 

#### 3.2.2. Alternative Splicing Takes Place in Gene Co-Expression Rather than Gene Deregulation Domains

Recently, we have addressed gene expression changes within the framework of spatial organization of up/downregulation [24] and gene co-expression [25], the latter applied on the same dataset presented herein. Based on the observation that AS events occur in genes that are distinct from the ones affected by DE, we explored their possible spatial co-occurrence in broader regions. 

We obtained domains of focal deregulation (DFD), as described in [24], and used the deduced domains of coordinated expression (DCE) from [25] to examine whether genes affected by AS tend to overlap with them to a greater extent than the one expected by chance. We performed a spatial overlap enrichment analysis implementing a permutation test as described in [26] and separately for various categories of DCEs and AS events (alternative acceptor/donor, exon skipping and intron retention). We found that AS events do not tend to overlap with domains enriched in deregulated genes, as suggested by the small overlaps of AS-affected genes with the DFDs (Figure 5, Appendix A). On the other hand, genes affected by AS, in particular by exon skipping and intron retention, appear to be enriched in chromosomal domains with increased gene co-expression (Figure 5, Appendix A). 

An additional trend in the association between AS and co-expression domains, was the increased enrichment in DCE regions that are dynamically affected by disease progression. DCEs that become disrupted in disease (reduced in size through splits and contraction) were shown to be more enriched in AS events compared to those that remained intact or become expanded. This observation may be linked to recent findings by our group on the same dataset [25], according to which, disease activity is correlated with extensive fragmentation and reorganization of gene co-expression. In all, the spatial distribution of AS genes corroborates their distinct behavior from over- or under-expressed genes, but is suggestive of their general enrichment in areas of increased regulatory dynamics, as represented by DCEs. 

### 3.3. Functional Enrichment of Alternative Splicing Suggests Previously Unreported Genes and Functions in the Context of SLE 

We next examined the functional characteristics of genes involved in AS events. We performed functional enrichment analysis with three different tools that focus on different aspects of gene categorization. The rationale behind this analysis was to examine the enrichment of AS-affected genes as members of particular gene ontology terms and/or as targets of particular transcriptional regulators. We thus examined the over-representation of AS-affected genes in gene ontology terms and in genes containing the binding site for transcription factors. A gProfileR [22] analysis was performed separately for those genes with significant changes in splicing (as assessed through dPSI) in inactive-vs.-healthy and active-vs.-healthy controls. The results for Gene Ontology terms and transcriptional regulators are shown in Figure 6. For the inactive disease state, we observed significant enrichments in the nucleoplasm and histone modification processes, in particular demethylation. Notable transcription factors that were enriched include Retinoblastoma (Rb), E2F and AP-2. 

Enrichments were much more pronounced in the comparison between active SLE patients and healthy controls, as the number of genes was also greater. Of note, we found a strong representation of terms related to cilium organization and assembly (Figure 6). At the level of transcriptional regulators, targets of the E2F family are also found to be enriched in this case. Of particular interest is the inclusion of the histone deacetylase HDAC2 in the list of the active state enriched regulators an enzyme which forms part of the REST repressor complex and exerts general repressive functions in various contexts. Finally, we employed Cluster Profiler [23] to assess the functional enrichment of AS-affected genes either in inactive or active patient samples. The results, shown in Figure 7, further support the observed enrichments for cilia formation and organization.

## 4. Discussion

Alternative splicing (AS) has long been suggested as a very important process in shaping the gene expression program. Disruption of AS in pathological conditions, especially in cancer, has also been widely reported. In this work, we performed a comprehensive computational exploration of AS in whole blood samples from a large cohort of SLE patients which enabled a quantitative assessment of AS events and a qualitative identification of genes and pathways affected by AS. 

One of the most interesting findings of this analysis is the prevalence of intron retention. Extensive intron retention is, in most cases, leading to disrupted open reading frames and thus non-viable mRNA molecules. It is thus mostly associated with gene repression and has been proposed as a mode of counter-balancing increased expression levels of undesirable transcripts. The predominance of intron retention events is noteworthy, considering that this type of alternative splicing outcome is the least common in mammals [11] but becomes prevalent under specific conditions such as developmental stages of hematopoiesis [13] or in certain types of cancer [27,28]. In these cases, intron retention, coupled with NMD may be seen as an additional mode of transcriptional regulation and, eventually, gene suppression. Intron retention events per se are difficult to assess as intronic transcription may be reflecting a general disruption in the dynamics of the transcription process. The large number of highly statistically significant such events in our dataset points towards a general trend that may imply specific functional tendencies. A number of test cases follow the pattern of *ACO1*, shown in Figure 8, where a considerable enrichment in intronic transcription, even exceeds the one in the exonic parts of the transcript. The strong intron retention patterns, combined with (a) the lack of over-expression among the affected genes, (b) the increase of dPSI for certain genes between inactive and active patient samples suggests that, AS in SLE is more likely to be implicated in the modulation of gene expression particularly towards repression. This is further supported by the prediction of transcripts leading to NMD. Whether this process is directly implicated to disease pathogenesis or represents a secondary (compensatory) effect remains to be investigated.

Another notable observation is the very small overlap between genes affected by AS and DE. This dissociation is not confined to individual genes but extends to broader genomic regions. The depletion of AS-affected genes in our defined domains of focal deregulation, suggests a spatial segregation of the two processes that take place in different sets of genes and within distinct chromosomal areas. Contrary to that, we find an enrichment of AS in regions of the genome with increased co-expression and, perhaps more importantly, in the subset of these regions that are more volatile when comparing healthy and diseased samples [25]. Together these findings further indicate the implication of AS in SLE in the generalized context of transcriptional dynamics. Moreover, these results suggest that the recently proposed domains of co-ordinated expression (DCE), identified on the same dataset [25], may indeed constitute genomic areas of increased regulatory activity and that alternative splicing is preferably taking place within their confines. In general, this finding is also interesting within the framework of spatial transcriptomics, which is becoming increasingly relevant in the context of complex diseases and SLE in particular.

At another level of AS-vs.-DE gene comparison, we found a very limited overlap of AS-affected genes between inactive and active SLE patients, when compared to healthy controls. When seen against the significant similarities of the differentially expressed genes in these states, this suggests that AS events could better discriminate active vs. inactive SLE, as compared to differential expression. SLE is a disease with a very wide range of physiological responses, but the molecular underpinnings of such breadth of activity remain yet largely undefined.

Additional indications of the dissociation between AS and DE come from the functional analysis of the corresponding gene sets, which point to different functions being enriched. Our analysis revealed enrichments of pathways and transcriptional regulators that are not directly linked to the inflammatory and immune responses. At a second level, though, we were able to identify genes associated with metabolic, epigenetic and cellular structure, all of which have been implicated in secondary immunological responses.

Functional enrichment analysis revealed that the protein HDAC2 targets genes that are subject to AS in SLE patients with high disease activity. HDAC2 is a histone modifying enzyme that catalyzes histone deacetylation, mediating transcriptional repression, but increasing evidence suggests it may also be associated with RNA splicing and interact with splicing factors [29]. Although the implication of HDAC2 in splicing has not been proven in the context of SLE, several studies indicate its role in inflammatory responses [30,31,32].

Among the various enriched functional terms, cilium organization is a rather unexpected set of functions given that cilia formation does not occur in lymphocytes [33,34]. Of interest, a recent study discussed the repurposing of cilia by T-cells [35], while several proteins sharing functions in both ciliated cells and CD4+ T cells had been already reviewed [36]. These proteins facilitate TCR and APC interaction and thus aberrant splicing of their genes could affect their expression and subsequently the immunological synapse.

Among particular gene cases, *ACO1* was the alternative spliced gene with the greater dPSI score between active and healthy samples (Figure 8). Turning back to the DE genes to see if there were functional enrichments in particular processes associated with ACO1, we found iron ion homeostasis and oxidative phosphorylation to be enriched (Appendix A), which is suggestive that ACO1 may have a potential effect in these processes through its repression. *ACO1* codes for the cytoplasmic isoform of cis-aconitase, an enzyme that catalyzes the isomerization of citrate to isocitrate, a part of the tricarboxylic acid cycle which is strongly linked to many inflammatory conditions, through the function of the decarboxylase IRG1, which is encoded by Interferon Responsive Gene 1 (*IRG1*), one of the most upregulated genes in most inflammatory conditions and is upregulated by interferon [37]. Citrate is also a key metabolite that affects the stoichiometric balance of Acetyl-CoA, through the action of ATP-citrate lyase. The latter has been shown to be upregulated in inflammation, while Acetyl-CoA is then incorporated into histone molecules in a process known as glucose-dependent histone acetylation.

In conclusion, our analysis is suggestive of a much more complex scheme of gene regulation in the context of SLE, while our results point towards a number of previously unreported genes and pathways as potential effectors of the disease. The large number of genes potentially affected by AS, their very small overlap with DE genes and their association with different functional processes and distinct areas of the genome, all suggest that AS could be a significant component of the gene regulation programme. The coupling of AS analysis with differential gene expression is bound to provide a more nuanced profile of the studied transcriptome.

## Figures and Tables

**Figure 1 genes-12-01260-f001:**
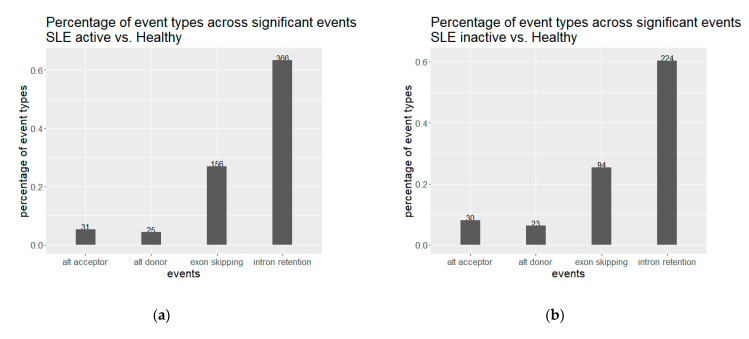
Alternative splicing events for (**a**) inactive and (**b**) active SLE patient samples against healthy controls. Intron retention is predominant in both comparisons.

**Figure 2 genes-12-01260-f002:**
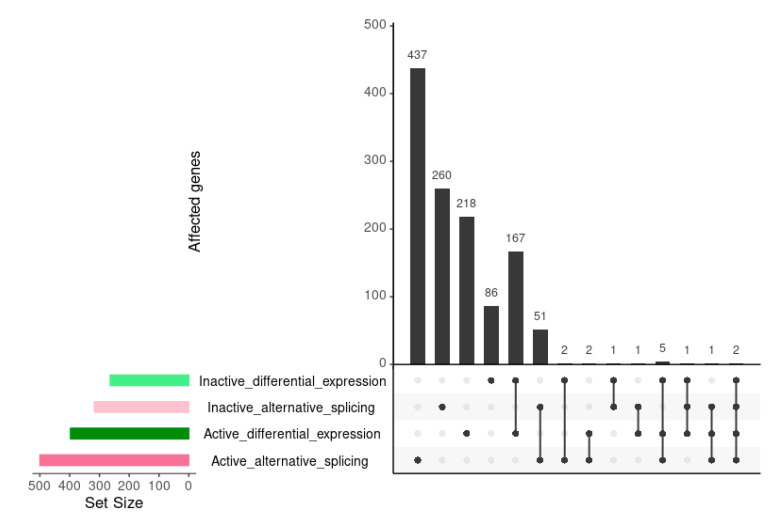
Upset comparison of genes affected by differential expression and alternative splicing in the inactive vs. healthy and the active vs. healthy comparisons. Horizontal bars show the number of genes belonging in each of the compared sets. Connected dots in the grid correspond to the combination of sets. Vertical bars show the number of genes shared among the sets in each combination.

**Figure 3 genes-12-01260-f003:**
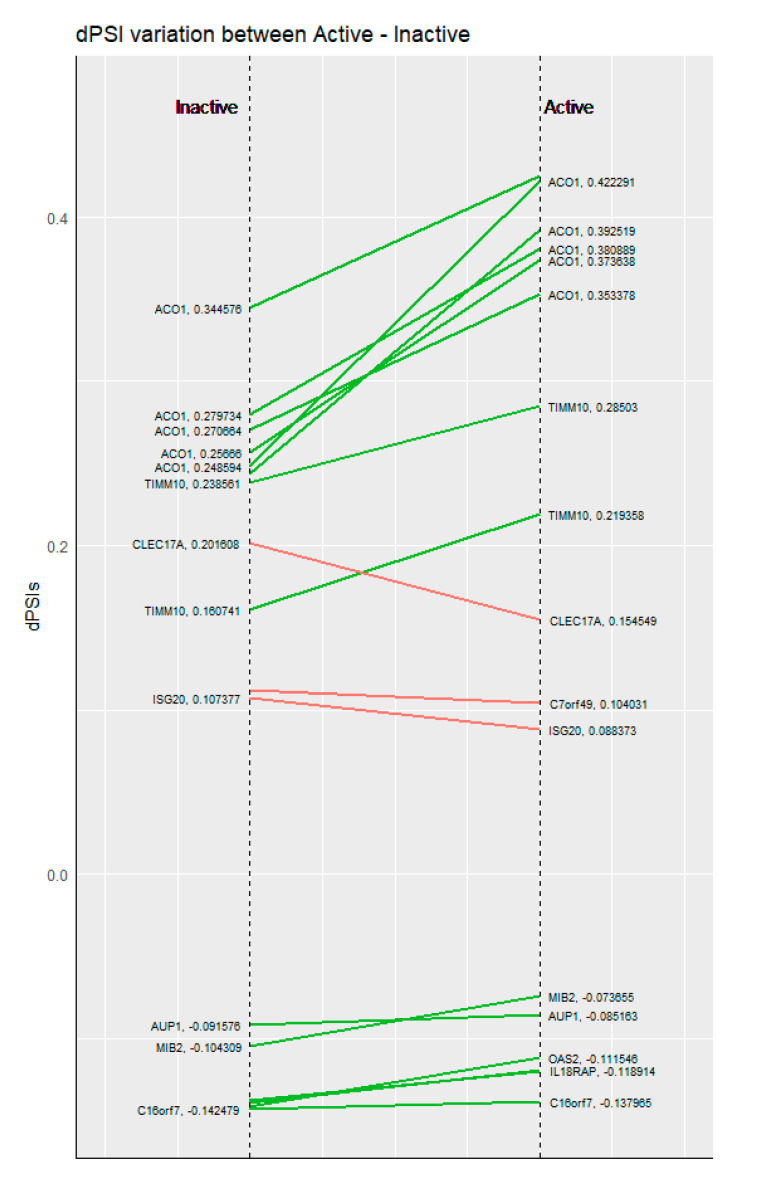
dPSI comparisons between inactive (**left**) and active (**right**) SLE patients against healthy controls, for the genes commonly affected by alternative splicing.

**Figure 4 genes-12-01260-f004:**
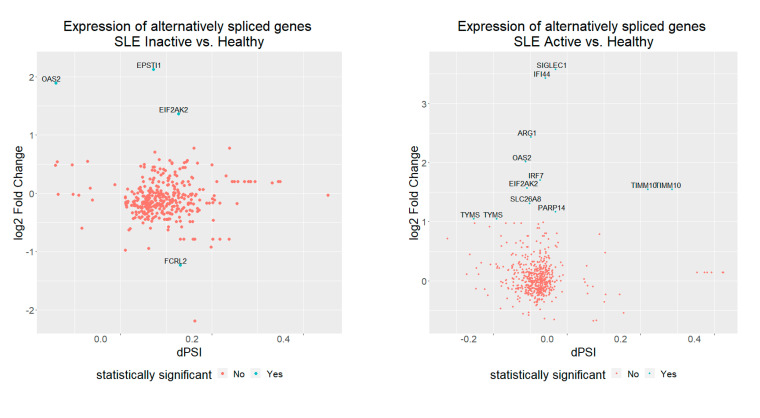
Alternative splicing, measured as dPSI score against differential expression, measured as log2-fold-change, for alternative spliced genes. Only a few genes (light blue) show differential expression, with the majority being over-expressed.

**Figure 5 genes-12-01260-f005:**
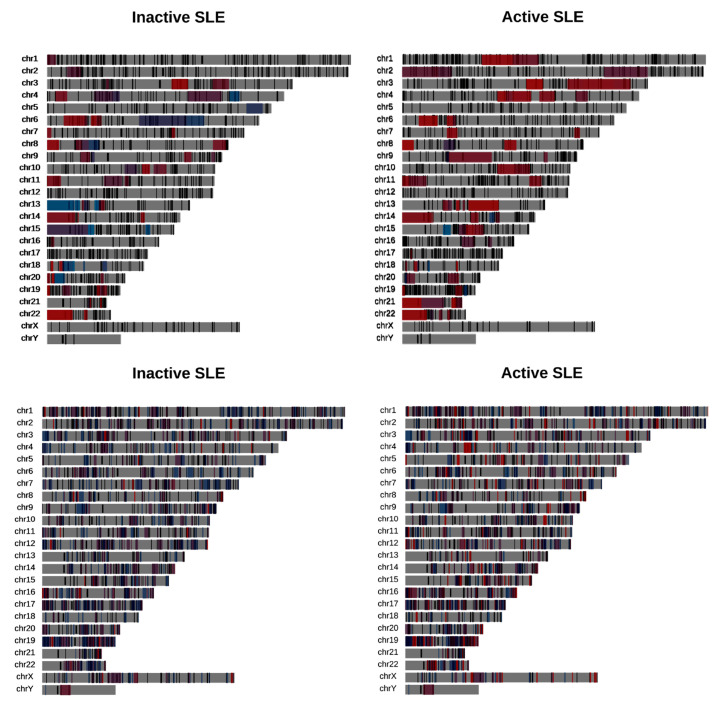
Overlaps between DFD (**top**) and DCE (**bottom**) and genes affected by AS in inactive (**left**) and active (**right**) SLE patient samples. **Top**: DFD color denotes down (blue) or upregulation (red). **Bottom**: DCE color denotes the DCE dynamics. Blue (contracted, depleted or split DCE), red (intact, merged or expanded DCE).

**Figure 6 genes-12-01260-f006:**
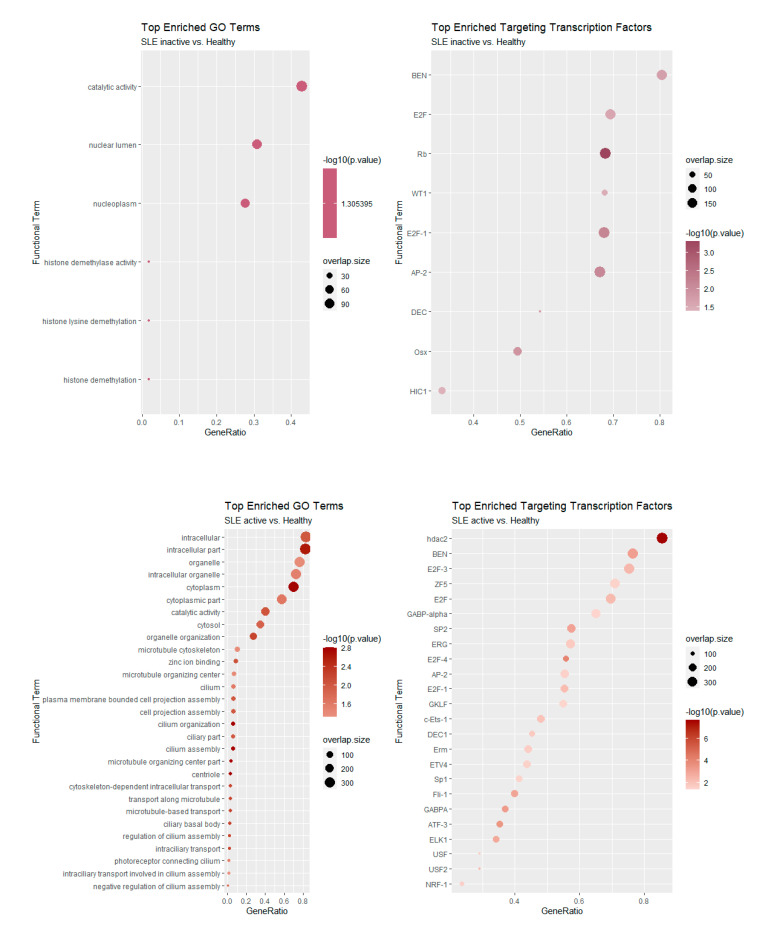
Functional enrichment of Gene Ontology terms (**left**) and transcriptional regulators (**right**) for inactive (**top**) and active (**bottom**) SLE patient samples. Analysis performed with gProfileR [22].

**Figure 7 genes-12-01260-f007:**
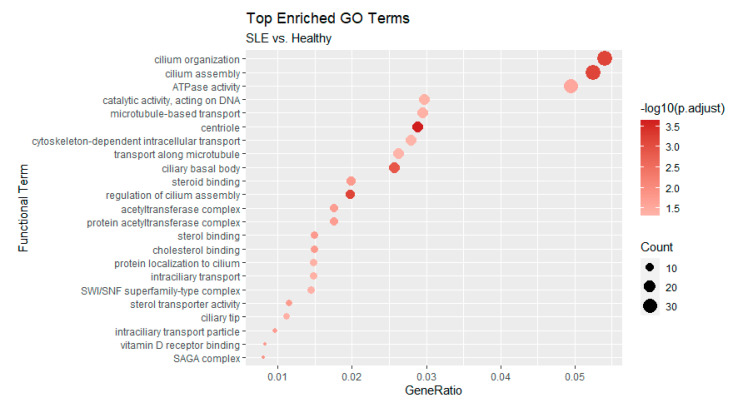
Functional enrichment of Gene Ontology terms based on AS affected genes that were found in inactive and/or active samples. Analysis performed with Cluster Profiler [23]. All *p*-values are adjusted at an FDR of 0.05.

**Figure 8 genes-12-01260-f008:**
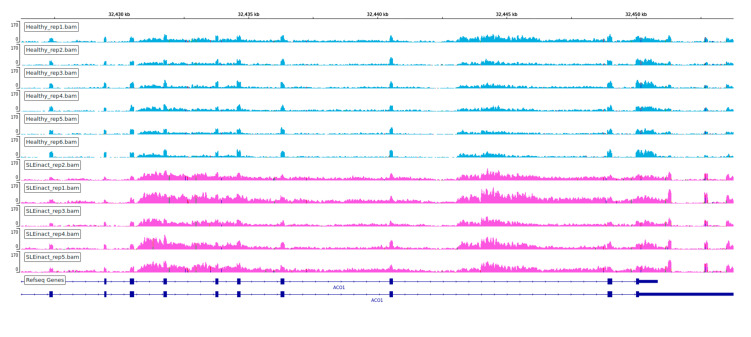
Visualization of coverage of each replicate for ACO1 gene. Multiple intron retentions found in SLE inactive (**top**, pink) and SLE active (**bottom**, red) samples.

**Table 1 genes-12-01260-t001:** Subgroups of RNASEq samples after merging.

State	Number of Samples	Number of Subgroups
SLE active	79	8
SLE inactive	46	5
Healthy	58	6

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
