# Peer review of "Extensive Changes in Transcription Dynamics Reflected on Alternative Splicing Events in Systemic Lupus Erythematosus Patients"

_genes, 2021, doi:10.3390/genes12081260_

Round 1

Reviewer 1 Report

The manuscript by Papanikolaou et al investigates splicing changes in SLE using a large set of patient vs healthy samples. This manuscript builds upon an already published study using the same RNAseq data that did not go into too much details regarding splicing alterations. This study attempts to do so but unfortunately does not provide a coherent analytical approach as suggested by the authors. Besides the initial splicing analysis, the authors perform many analysis that do not seem to fit together towards a reasonable conclusion. In fact some of these analyses seem to provide contradictory data that the authors failed to address.

Specific comments:

The authors should provide some experimental confirmation to several of the splicing alterations detected by the bioinformatic approach. The whole manuscript is based on computational analyses without any lab-based conformations.

The authors investigated the introduction of PTCs as a result of intron retention and the potential elimination of transcripts via NMD. They found >90% of retained intron transcripts can be subjected to elimination via NMD. They actually make this a central point of their manuscript. If it indeed occurs, NMD which is described as a way to suppress gene expression would have a clear effect on transcript levels and should be detected by the DE analysis. Yet there is little to no overlap between genes that show retained introns and differential expression (specifically downregulation).

In Figure 6, the authors present data on transcriptional regulators of genes that show alternative splicing. This reviewer is baffled by this analysis as there is no reasonable link between transcription regulation and splicing in this particular manuscript.

The discussion/explanation for "a very limited overlap of AS-affected genes between inactive and active SLE patients, when compared to healthy 350 controls" is vague and not clear. 

The authors mentioned that they merged the samples into multiple subgroups to increase depth. were these samples randomly merged?

Why did the authors align their input files to an outdated hg19 rather than hg38?

The panels in Figure 1 are duplicated.

this reviewer does not get the value of the analyses done in Figure 5, and Table 3. They do not seem to add much to our understanding of the role of alternative splicing in SLE.

Author Response

Dear reviewer,

We appreciate your thorough and timely review of our manuscript. We have read them carefully and have tried to incorporate the majority of your suggestions.

We believe that we revised version of our work is greatly improved on the basis of your criticism.

Please find our responses to your comments in bold below.

===

Comments and Suggestions for Authors

The manuscript by Papanikolaou et al investigates splicing changes in SLE using a large set of patient vs healthy samples. This manuscript builds upon an already published study using the same RNAseq data that did not go into too much details regarding splicing alterations. This study attempts to do so but unfortunately does not provide a coherent analytical approach as suggested by the authors. Besides the initial splicing analysis, the authors perform many analysis that do not seem to fit together towards a reasonable conclusion. In fact some of these analyses seem to provide contradictory data that the authors failed to address.

Specific comments:

The authors should provide some experimental confirmation to several of the splicing alterations detected by the bioinformatic approach. The whole manuscript is based on computational analyses without any lab-based conformations.

We appreciate this comment, however this was meant by design to be a purely computational work. Our intentions were to re-visit an already published set of data in order to showcase a) the potential of standard RNASeq profiling in providing information on splicing events b) the importance of the latter in achieving a more concise view of transcription dynamics. Small scale experimental validation is therefore beyond the scope of the proposed article. We discuss this point in the manuscript (Discussion, page 11).

The authors investigated the introduction of PTCs as a result of intron retention and the potential elimination of transcripts via NMD. They found >90% of retained intron transcripts can be subjected to elimination via NMD. They actually make this a central point of their manuscript. If it indeed occurs, NMD which is described as a way to suppress gene expression would have a clear effect on transcript levels and should be detected by the DE analysis. Yet there is little to no overlap between genes that show retained introns and differential expression (specifically downregulation).

NMD is a translation-related regulation process and in this sense, it may affect mature mRNAs that are bound by ribosomes. Accordingly, NMD could act as a balancing mechanism to readjust or even neutralize increased mRNA levels. A more careful look at the data we present may lead to a plausible scenario, according to which pre-mRNA of certain transcripts is stimulated by a given signal, this leads to increased overall transcript levels, which are subsequently depleted via NMD, triggered by intron retentions. The combination of the two processes will cancel out the effect and therefore the gene will not be recorded as down-regulated. One interesting point in favour of this scenario is that some of our most striking examples (such as ACO1) are upregulated in inflammatory conditions. Our data suggest that intron retention-mediated NMD may neutralize this effect (see discussion on Figure 4).

We have now tried to make this argument clearer in the revised manuscript.

In Figure 6, the authors present data on transcriptional regulators of genes that show alternative splicing. This reviewer is baffled by this analysis as there is no reasonable link between transcription regulation and splicing in this particular manuscript.

One of the major points in this work, is that splicing is probably acting as a balancing process to counteract increased mRNA levels (see response in previous comment). The point behind the regulatory enrichment analysis is to identify exactly those regulators that may drive increased mRNA expression that is then balanced by splicing.

We have also included discussion on this argument in the revised manuscript.

The discussion/explanation for "a very limited overlap of AS-affected genes between inactive and active SLE patients, when compared to healthy 350 controls" is vague and not clear. 

We thank the reviewer for pointing this and other parts of the manuscript that may, indeed, be unclear. We could not find the particular quote mentioned above. but we have nonetheless taken this as a general point into consideration in an updated version of the manuscript.

The authors mentioned that they merged the samples into multiple subgroups to increase depth. were these samples randomly merged?

Yes, samples were randomly merged. We have added this clarification in the manuscript.

Why did the authors align their input files to an outdated hg19 rather than hg38?

The analysis we present is based on a RNASeq dataset that we previously published (Panoussis et al, 2019), in which the hg19 assembly was used. We opted for the same version in order to be able to cross-correlate our findings in a more direct manner. Even though the hg38 is more complete, the differences with hg19 are mostly confined in SNPs and indels from non-coding (non-intronic) regions.

The panels in Figure 1 are duplicated.

In our version of the article the panels are OK. However, we will make sure it is clearly visible in the revised version of the manuscript.

this reviewer does not get the value of the analyses done in Figure 5, and Table 3. They do not seem to add much to our understanding of the role of alternative splicing in SLE.

We thank the reviewer for this point. We do agree that Tables 2 (and perhaps also Table 3) do not provide dense information and have now moved them to the supplementary material. However, the rationale behind Figure 5 is to showcase one crucial aspect of the relationship between AS and transcriptional dynamics. What Figure 5 shows is that AS events tend to occur not in areas of differential expression clustering but in genomic regions where co-expression between patients of the same activity tends to be more pronounced. We believe this to be indicative of an interesting aspect of AS, a sort of segregation of functions with de-regulation occurring in some areas of the genome and AS mostly taking place elsewhere. We therefore ask the reviewer to examine Figure 5 under this prospect of spatial transcriptomics.

We have also tried to clarify the above point in the corresponding part of the manuscript.

Reviewer 2 Report

The manuscript by Sofia Papanikolaou et al. tries to identify a functional link between aberrant AS events and gene categories that could be of potential interest for SLE. Focusing on the biological meaning, and not on the technical approach, I found the manuscript of potential interest. Unfortunately, no significant correlations between AS and gene expression came out from this study, but I believe that this kind of analysis should be encouraged to identify new molecular mechanisms at the bases of human diseases. I have some concerns that I hope will be addressed by the Authors.

The main criticism is related to ACO1 and the conclusions the authors drew. The authors focused on just 1 of the very few genes identified as deregulated in AS events and try to find a correlation between ACO1 and DE genes in SLE. GSEA identified iron homeostasis and oxidative phosphorylation to be enriched in SLE vs healthy samples, but I don’t find this simple correlation strong enough to ascribe to ACO1 AS a key role in SLE (this result is overstated as highlighted by the title of the manuscript itself). It is not clear how altered AS in ACO1 identified in active SLE can affect these transcriptional pathways. Can the authors better explain how altered AS of ACO1 affects the abundance of specific transcripts and translated proteins and consequently on metabolic pathways? Are the different AS transcript described to have different functional activities? If the authors decide to point their attention on ACO1 I suggest to add key details on the function of its isoforms and validate their results at least by qRT-PCRs in active SLE patients (or to retrieve these data from RNA-seq data).

To better understand the presented results, I find relevant to add a brief description of the parameters through which patients have been classified as active or inactive SLE.

Figure 1: Most of the results described in the first paragraph (3.1) are not presented in Figure 1. Is it possible to add statistical information to better evaluate the difference in absolute ΔPSI scores in active SLE patient samples and inactive ones?

Figure 2: it should be useful a detailed explanation of the figure in the legend

Line 226: what do the author mean with “Notable transcription factors that were enriched include Retinoblastoma (Rb), E2F and AP-2”? Are these the transcription factors involved in transcriptional regulations of genes affected by AS? If so, which functional link exist between transcription factors that bind to promoter regions and altered AS events? It is not clear to me the rationale of this analysis, which would be appropriate if DE was examined in place of AS.  I think it should be more informative to identify whether AS events deregulated in SLE are associated to deregulation of specific splicing factors rather than transcription factors. It could be that specific splicing factors are involved as a common mechanism of action as described in other human diseases. Is it possible to modify Figure 6 similarly to Figure 7? The enriched GO categories are represented with circles of the same dimensions. If possible, I suggest to adjust the size of each circle on the number of genes included in each GO term and possibly the color on the adjusted pvalue.

Please, define abbreviations at first mention and avoid abbreviations in the abstract: DE (abstract), IR (line 52)

Author Response

Dear reviewer,

We appreciate your thorough and timely review of our manuscript. We have read them carefully and have tried to incorporate the majority of your suggestions.

We believe that we revised version of our work is greatly improved on the basis of your criticism.

Please find our responses to your comments in bold below.

===

The manuscript by Sofia Papanikolaou et al. tries to identify a functional link between aberrant AS events and gene categories that could be of potential interest for SLE. Focusing on the biological meaning, and not on the technical approach, I found the manuscript of potential interest. Unfortunately, no significant correlations between AS and gene expression came out from this study, but I believe that this kind of analysis should be encouraged to identify new molecular mechanisms at the bases of human diseases. I have some concerns that I hope will be addressed by the Authors.

The main criticism is related to ACO1 and the conclusions the authors drew. The authors focused on just 1 of the very few genes identified as deregulated in AS events and try to find a correlation between ACO1 and DE genes in SLE. GSEA identified iron homeostasis and oxidative phosphorylation to be enriched in SLE vs healthy samples, but I don’t find this simple correlation strong enough to ascribe to ACO1 AS a key role in SLE (this result is overstated as highlighted by the title of the manuscript itself). It is not clear how altered AS in ACO1 identified in active SLE can affect these transcriptional pathways. Can the authors better explain how altered AS of ACO1 affects the abundance of specific transcripts and translated proteins and consequently on metabolic pathways? Are the different AS transcript described to have different functional activities? If the authors decide to point their attention on ACO1 I suggest to add key details on the function of its isoforms and validate their results at least by qRT-PCRs in active SLE patients (or to retrieve these data from RNA-seq data).

We thank the reviewer for this constructive criticism. We agree that some of our data may have been overstated in the title and inside the text. They are grounded on preliminary, unpublished data from other lines of research in our group as well as on increasing body of evidence in already published works. We have now a) downplayed the importance of ACO1 in changing the title and b) rewritten the corresponding parts of the discussion.

Experimental validation, even though within our prospective goals, is beyond the scope of the proposed work and would require a significantly extended deadline for submission of the revised manuscript.

To better understand the presented results, I find relevant to add a brief description of the parameters through which patients have been classified as active or inactive SLE.

These have been extensively described in the cited previous works from our labs (Panousis et al., 2019; Ntasis et al., 2020). We have now added a brief discussion for more clarity.

Figure 1: Most of the results described in the first paragraph (3.1) are not presented in Figure 1. Is it possible to add statistical information to better evaluate the difference in absolute ΔPSI scores in active SLE patient samples and inactive ones?

As dPSI scores are not directly comparable except when discussing the same event, we have now removed Figure 1b as we realize that it was causing confusion and was not providing any useful information. We thank the reviewer for pointing this out.

Figure 2: it should be useful a detailed explanation of the figure in the legend

An updated version of the legend explains the plot in greater detail.

Line 226: what do the author mean with “Notable transcription factors that were enriched include Retinoblastoma (Rb), E2F and AP-2”? Are these the transcription factors involved in transcriptional regulations of genes affected by AS? If so, which functional link exist between transcription factors that bind to promoter regions and altered AS events? It is not clear to me the rationale of this analysis, which would be appropriate if DE was examined in place of AS.  I think it should be more informative to identify whether AS events deregulated in SLE are associated to deregulation of specific splicing factors rather than transcription factors. It could be that specific splicing factors are involved as a common mechanism of action as described in other human diseases. Is it possible to modify Figure 6 similarly to Figure 7? The enriched GO categories are represented with circles of the same dimensions. If possible, I suggest to adjust the size of each circle on the number of genes included in each GO term and possibly the color on the adjusted pvalue.

This is indeed an interesting point. Given the extensively reported intron retention and the likelihood of NMD, the rationale behind the transcriptional regulators analysis is to examine possible links between genes that would be expected to be overexpressed but whose levels are balanced by AS. This question was raised by another independent reviewer and so we have tried to explain our motivation for this analysis in clearer way in the revised version of the text.

We have performed an analysis for splicing factors but could not find any significant enrichments.

In additions, figures 6 and 7 were created with two different software and are therefore produced differently by default. We have now modified our code according to the suggestion of the reviewer in order to present additional information on Gene Ratio, number of genes in term and adjusted p-value.

Please, define abbreviations at first mention and avoid abbreviations in the abstract: DE (abstract), IR (line 52)

We thank the reviewer for this comment. We have modified the text accordingly.

Reviewer 3 Report

Review on the manuscript titled “Extensive intron retention regulates the expression of meta-2bolic and DNA modifying genes in Systemic LupusErythema-3tosus patients” by Papanikolaou et al., 2021.

         The authors observed massive aberrant intron retention (IR) events in metabolic and DNA modifying genes in SLE patients compared to the control. As a conclusion, the authors underscored the importance of Alternative splicing SLE.

While VastDB/Vastools implies the majority of AS events are IR (PMID: 28855263; 2017; Fig.1), there is a long standing fact that IR in animals are the least frequent events, in particular due to hughe intron sizes (McGuire AM, Pearson MD, Neafsey DE, Galagan JE. Cross-kingdom patterns of alternative splicing and splice recognition. Genome Biol. 2008;9(3):R50. PMID: 18321378; PMCID: PMC2397502).

Secondly, the same time all RNA-Seq datasets contain partial intron RNA sequences. In particular the term ‘RNA velocity’ used in single cells data clustering based on evaluation of unsplicied/spliced transcripts ratio yielding overall quantitative dynamics of the transcriptomes (La Manno et al. RNA velocity of single cells. Nature. 2018;560(7719):494-498. PMID: 30089906; PMCID: PMC6130801). Thus, while it’s known that plenty of sequenced RNA-seq data display unprocessed/partially processed mRNA, it is mostly due to  temporal  nature of splicing routine, and proving to be informative, but not considered as IR event  (PMID: 30089906).

Thirdly, the use of exon-intron-exon trios junctions coverage cannot be treated the same way for IR elucidation as for Exon skipping events due to outstanding intron length, unless few cases it’s a really small one.

Fourthly, after a prolonged round of AS RNA-Seq analysis from 2014 up to now, a paper devoted to IR recognition challenge appeared, claiming poor accuracy and quite complicated procedure for IR detection in RNA-Seq analysis, dated 2020 (PMID: 32206209), which I consider crucial addressing for IR detection issues. Notably, its authors didn’t list VastTools as IR identification program suitable for the robust assessing of this event in their AS software list.

Thus, while the issue of intron retention is quite interesting, there are profound problems in correct identification and evaluation of such events.

Most of the alternative splicing software packages widely applicable for RNA-Seq data AS analysis don’t assess IR splicing mode due to the difficulties of its robust identification listed above.

Overall, I suggest the authors trying some other AS tools (rMAts, etc) for assessing AS for being compliant based on several tools. Alternatively, if the authors insist on their results, they should provide explicit cross-control procedures addressing the issues listed below on IR detection from the publication mentioned, in particular:

“The accurate detection of retained introns and precise measurement of intronic expression are crucial to these studies. Numerous factors impede the detection of IR from next generation sequencing data. Introns are much longer than exons and thus have a much higher probability of containing overlapping features that may confound the estimation of intronic expression. In addition, introns are enriched in low complexity and repeat sequences that may prevent sequencing data from being uniquely mapped. These factors must be accounted for when detecting IR events. Most computational approaches however will introduce a selection bias as only introns with sufficient coverage can be detected and the statistical power required to detect differences between conditions increases with coverage depth and the read count [48], [49]. As a result of this bias, gene enrichment tests of genes derived from IR signatures [50] are heavily skewed towards the more expressed genes and towards introns that do not contain these confounding features.

Despite the recent results that demonstrate a crucial role for IR, very few IR events have been validated in the wetlab and amongst these an even smaller portion have been investigated for their functional impact. As a consequence, no reliable benchmark of IR detection or differential intronic expression has been published. This lack of reliable controls is however temporary because long read technologies capable of sequencing entire IR transcripts help resolve most of the detection problems. However, due to their low coverage, these technologies are far from allowing a comprehensive detection of IR events and even further from allowing a reliable quantification of IR levels between different tissues.” (PMID: 32206209).

    I state that IR events reported are mostly false positives unless proved otherwise.

Author Response

Dear reviewer,

We appreciate your thorough and timely review of our work. We have tried to incorprorate the majority of your suggestions in what we believe to be an improved version of our manuscript.

Please find our detailed point-by-point responses in the attached file. 

Round 2

Reviewer 2 Report

I thank the authors for having addressed my issues.

However, I recommend the authors to address the following minor comments.

  1. Figure 1: match dimension and font size of the two panels. Results 3.1, line 10: Figure S1 has been cited but does not represent what described in the text (S1 showed GSEA that now is not cited in the text and should be removed).
  2. Results 3.2.1, line 3: add the Reference demonstrating that “active SLE patients show a greater number and more intense changes at transcript levels”.
  3. Table 2: it would be very useful for readers to add a column with the transcription factor name near the Ensembl code. I’m sure that this would widen the viewing of the manuscript from scientists that can identify the transcription factor of interest for their research activity within this database.
  4. Results 3.3, line 10: I’m sorry but I still don’t understand the rationale behind the identification of transcription factors (TFs). What the Authors mean with “transcription factors were enriched” and “Top Enriched Targeting transcription factors”? Are these transcription factors (TFs) retrieved among transcriptionally up-regulated genes in the two groups? Or are these TFs identified among enriched TF binding sites in regulatory regions of genes transcriptionally deregulated in the two groups? The Authors described the results as the analysis of genes affected by AS for Gene Onthology terms and transcriptional regulators. From this description, it seems that TFs identified are retrieved from genes affected by AS.

Author Response

Response to Reviewer 1 Comments

Point 1: Figure 1: match dimension and font size of the two panels. Results 3.1, line 10: Figure S1 has been cited but does not represent what described in the text (S1 showed GSEA that now is not cited in the text and should be removed).

Response 1: We thank the reviewer for pointing this out. We have replaced Figure 1 with a new version. Regarding the Supplementary Figures we would like to point out that, in our more recent version of the manuscript, there are two Supplementary Figures which are correctly cited in the text. We urge the reviewer to make sure the editor has made this version available to her/him.

Point 2: Results 3.2.1, line 3: add the Reference demonstrating that “active SLE patients show a greater number and more intense changes at transcript levels”.

Response 2: We have now added this reference.

Point 3: Table 2: it would be very useful for readers to add a column with the transcription factor name near the Ensembl code. I’m sure that this would widen the viewing of the manuscript from scientists that can identify the transcription factor of interest for their research activity within this database.

Response 3: Again we have trouble corresponding the comment to the content of our manuscript. In our latest version, there is no Table 2. Transcription factor analysis is shown as part of Figure 6, in which the names of the transcription factors are shown in full.

Point 4: Results 3.3, line 10: I’m sorry but I still don’t understand the rationale behind the identification of transcription factors (TFs). What the Authors mean with “transcription factors were enriched” and “Top Enriched Targeting transcription factors”? Are these transcription factors (TFs) retrieved among transcriptionally up-regulated genes in the two groups? Or are these TFs identified among enriched TF binding sites in regulatory regions of genes transcriptionally deregulated in the two groups? The Authors described the results as the analysis of genes affected by AS for Gene Onthology terms and transcriptional regulators. From this description, it seems that TFs identified are retrieved from genes affected by AS.

Response 4: We thank the reviewer for pointing out this part of the manuscript where the analysis is not clearly described. The functional analysis has been indeed performed on AS genes. We have now tried to make this point clear in the text.

Reviewer 3 Report

One minor thing I'd like to ask presenting in the manuscript is the table of intron length distribution for IR events.

Author Response

Response to Reviewer 3 Comments

Point 1: One minor thing I'd like to ask presenting in the manuscript is the table of intron length distribution for IR events.

 Response 1: As it is difficult to provide the information for length distributions in a table we point the reviewer to Supplementary Figure 1, in which the length distribution for each category of events is being shown and to Supplementary File 1, in which the length of each significant AS even is being incorporated as a final column.